# The Role of Geriatric Assessment in the Management of Diffuse Large B-Cell Lymphoma

**DOI:** 10.3390/cancers15245845

**Published:** 2023-12-14

**Authors:** Francesco Merli, Stefano Pozzi, Hillary Catellani, Emiliano Barbieri, Stefano Luminari

**Affiliations:** 1Hematology Unit, Azienda USL-IRCCS di Reggio Emilia, 42123 Reggio Emilia, Italy; stefano.luminari@ausl.re.it; 2Clinical and Experimental Medicine PhD Program, University of Modena and Reggio Emilia, 41121 Modena, Italy; stefano.pozzi@ausl.re.it (S.P.);

**Keywords:** diffuse large B-cell lymphoma, simplified geriatric assessment, older patients

## Abstract

**Simple Summary:**

The treatment choice for an older patient with diffuse large B-cell lymphoma (DLBCL) is challenging due to the complexity of the patient. Among the several available tools to evaluate an older subject with DLBCL, the simplified geriatric assessment (sGA) categorizes patients as fit, unfit, or frail and has been validated to predict the risk of death. The elderly prognostic index (EPI), which combines sGA and IPI scores and hemoglobin level, is the first prognostic score for older patients, with three risk groups for survival. New validated tools will help physicians choose the best treatment for elderly patients with DLBCL, further contributing to improving the personalized approach to elderly subjects.

**Abstract:**

The treatment choice for an older patient with diffuse large B-cell lymphoma (DLBCL) depends on many other factors in addition to age, which alone does not reflect the complexity of the aging process. Functional features and comorbidity incidence differ not only between younger and older patients but also among older patients themselves. The comprehensive geriatric assessment (CGA) quickly evaluates fitness status by investigating the patient’s different functional areas, degree of autonomy, and presence of comorbidities. Various tools are available to evaluate frailty; which assessment tool to use should be based on the clinical aim. The simplified geriatric assessment (sGA) from the elderly project by the Fondazione Italiana Linfomi, prospectively tested on the largest number of patients, categorizes patients as fit, unfit, or frail, with a decreasing rate of overall survival. The elderly prognostic index (EPI), which combines sGA and IPI scores and hemoglobin level, is the first prognostic score for older patients, with three risk groups for survival. Future GAs should consider new parameters, including sarcopenia, which appears to be inversely related to survival. New tools based on prospective studies can help physicians choose the best treatment in light of the individual patient’s characteristics.

## 1. Introduction

Diffuse large B-cell lymphoma (DLBCL) is the most common subtype of non-Hodgkin lymphoma [1], with the median age of patients exceeding 65 years [2] and with approximately one-third of cases who are over 75 years [3]. Of note, the elderly population is estimated to increase significantly in the next few years [4], with a concomitant expected growth in the absolute number of DLBCL patients [5,6]. Even if DLBCL in the elderly population is confirmed as a curable cancer, treatment of older lymphoma patients is challenging.

R-CHOP represents the reference regimen in the first-line treatment algorithm for DLBCL, with an overall cure rate of roughly 60–70% [7]. Attempts have been made to improve R-CHOP efficacy by adding novel agents, including lenalidomide, ibrutinib, and bortezomib, but with no meaningful results thus far; results from ongoing trials are awaited [8,9,10]. The emergence of drug-conjugated antibodies [11,12] and the positive results achieved with the POLARIX phase III trial [13] have demonstrated that improving the efficacy of standard immunochemotherapy is possible by adding the antibody drug conjugate polatuzumab vedotin to R-CHP (pola-R-CHP). Full-dose anthracycline-based regimens are strongly supported by the results of well-designed phase III trials, which, however, enrolled patients until the age of 80 years and whose strict eligibility criteria resulted in the underrepresentation of older patients [14]. Thus, the evidence generated for younger patients cannot be generalized to older patients, for whom dedicated clinical trials are lacking. Several factors have limited the development of new treatments for older patients with DLBCL; in addition to their age, a critical prognostic factor, older patients accumulate a considerable number of unfavorable biological features [15,16,17] and geriatric syndromes, as well as a reduced tolerance to treatment. All of these can negatively affect the life expectancy of these subjects [18], independently of the lymphoma, but they do contribute to the complexity of therapeutic decision-making in this population. The main challenge for the treating physician is to evaluate the patient and the lymphoma to decide whether a cure is a feasible option for that patient and how to achieve this goal with the limited therapeutic options available. With no validated, objective tools to evaluate the patient and the disease, the therapeutic decision is left to the subjective assessment of the physician, with a possible risk of exposing the patient to unnecessary toxicity or to ineffective therapies. In this article, we review the main tools that can be used to support the initial assessment of the older subject and how these tools can be used to foresee DLBCL-patient outcomes and to support treatment decisions.

## 2. Tools to Evaluate Functional Status

Chronological age alone does not reflect the complexity of the aging process [19]. In addition to age, functional features, comorbidity incidence, and immunocompetence status differ greatly not only between younger and older patients but also between one elderly individual and another [20].

In addition to age thresholds, the ECOG performance status (PS) measure is frequently used to define a patient’s fitness status [21]. However, a recent review of geriatric assessment (GA) in 44 studies of hematological malignancies emphasized the inadequacy of PS [22]. While the overall median proportion of patients with at least one geriatric impairment was 51%, PS > 2 accounted for a median rate of only 29%, underestimating frailty status by about 20%. More importantly, the use of GA to detect geriatric impairments is better correlated with survival compared to the use of PS [22]. Moreover, PS may be strictly dependent on lymphoma-related symptoms at disease onset that can rapidly resolve after pre-phase therapy [23]. Conversely and differently from PS, activities of daily living (ADL) [24] and instrumental activities of daily living (IADL) [25] scores focus on a patient’s need to perform daily activities and to solve problems related to independent everyday life (i.e., ability to use the telephone, shop, prepare food, do housekeeping, etc.).

Although GA was better at assessing fitness status, most of the available clinical trials on the elderly have been based on age and PS. The use of specific comorbidity measures would help to improve the quality of clinical trials and would better support the therapeutic decision-making process in older patients. In this respect, the National Institute of Aging/National Cancer Institute’s Comorbidity Index was used in a population study of patients with aggressive lymphoma, showing the better prognostic role of comorbidities compared to the international prognostic index (IPI) [26,27,28]. The EORTC cooperative group has used the cumulative illness rating scale (CIRS) to estimate fitness status in the context of prospective studies on frail DLBCL patients [29]; severe comorbidities were identified in 37–47% of patients and were thus included to define frailty status.

The choice of which tool to use to measure frailty should be based on the clinical aim. The descriptions and areas of application of the above-mentioned tools or scales were summarized by Goede et al. [30]. Specifically, the 19-item Charlson comorbidity index (CCI) [31], the 14-item CIRS [32,33], and the less complicated 9-item clinical frailty scale (CFS) [34] adequately record frailty status pre-treatment and can be used to plan any necessary geriatric intervention. More complex assessments aim to predict clinical outcomes, record and plan actions, and measure their beneficial effects.

ADL and IADL have been validated in DLBCL studies. Patients with poor ADL were 4% in the two Tucci et al. studies [35,36], while they were 7%, 18%, and 27%, respectively, in the studies conducted by Ong et al. [37], Merli et al. [38], and Spina et al. [39]. In addition, a systematic review by Scheepers et al. confirmed the significant association between these scores and the likelihood of dying [22].

### 2.1. Comprehensive Geriatric Assessment (CGA) in DLBCL

The idea of functionally categorizing aging DLBCL patients based on their fitness status was first adopted in 2000 by Balducci & Extermann [40]. The authors tried to answer questions dealing with the substantial difference between dying with or because of cancer and with a patient’s capability to tolerate chemotherapy and its consequent risk-benefit ratio. These authors were the first to explore a multidimensional assessment of the elderly. Subsequently, different methods for CGA have been suggested, from the critical involvement of geriatricians to self-evaluation using single essential items [41].

Among the available studies Hormigo-Sanchez et al. recently demonstrated that standardized, systematic CGA performed by geriatricians permits classification of lymphoma patients by level of frailty, helps in decision making, and predicts clinical outcomes [42].

The Italian experience with CGA started in 2009 with the first monocentric contribution published by Tucci et al. [35], in which 84 patients (aged >65 years) were classified as fit, unfit, or frail based on age (cutoff = 80 years), ADL and CIRS-G scores, and the geriatric syndrome as defined by Balducci & Beghe [43]. IPI scored all cases as either intermediate high or high risk. As expected, full-dose-treated cases experienced significantly longer overall survival (OS) than those who received a palliative approach per the physician’s clinical judgment. Moreover, no substantial differences were found between CGA-fit patients treated with curative intent and younger patients who underwent the same treatment. On the other hand, unfit cases had poorer outcomes regardless of the type of treatment, suggesting that not only disease-associated features affect outcome but CGA-detected patient-associated features do as well [35].

Following Tucci et al.’s 2015 paper [36], the Fondazione Italiana Linfomi (FIL) conducted a first multicenter prospective study to validate those results in a larger population of older DLBCL patients (>69 years). The results of this prospective study confirmed the superior OS rate of fit patients (2-year OS rate = 88%) treated with curative intent compared with those non-fit cases receiving full-dose therapy (2-year OS rate = 47%), highlighting the therapy-independent predictive power of CGA. Examining the OS results from the single CGA categories, it is interesting to note that the curative approach significantly improved the 2-year OS of fit patients (88% vs. 25%; *p* < 0.0001), while a non-significant superior OS rate in both unfit (75% vs. 45%) and frail (44% vs. 39%) patients was found. Finally, multivariate analysis indicated that IPI and CGA scores were independently associated with OS. Finally, the survival of the CGA-fit patients was significantly better than that of patients treated with full-dose therapy as prescribed by the treating physician.

### 2.2. The Simplified Geriatric Assessment (sGA) and the Elderly Prognostic Index (EPI)

The elderly project (EP), a multicenter observational study conducted in 36 FIL centers, aimed to assess CGA’s influence on OS in newly diagnosed and prospectively followed-up elderly DLBCL patients. One of the considerable strengths of this prospective study was the large number of patients (1353 cases) registered from December 2013 to December 2017 [44]. The authors developed a new simplified GA (sGA), which was derived from the original CGA, so as to better assess the independent prognostic influences of age and fitness status. Notably, 24% of the original CGA cases were reallocated in the simplified assessment.

Table 1 shows the sGA items (Table 1). More than half of the cases were classified as fit, and 28% and 18% as unfit and frail by sGA scores, respectively, with a significantly different OS for the three sGA groups. Indeed, unfit and frail patients showed roughly a 2- and 3-times higher risk of dying, respectively, compared to fit patients. The 1.7 fold higher risk of mortality for frail versus unfit patients was also highly significant (Figure 1). Thus, the first remarkable result of this study is that, by validating the sGA, the authors operationally answered the question of what the best and currently most straightforward tool is to determine the prognosis of older DLBCL patients. Unlike a complete geriatric assessment, physicians can easily perform the sGA in just a few minutes; the calculator provided on the FIL website (www.filinf.it/epi, accessed on 29 October) is a powerful, non-time-consuming tool. A similar approach to rapidly establishing fitness using other specific selection tools has been adopted by other groups [45,46,47].

Importantly, the FIL group devised and validated a new prognostic model that integrates patient- and lymphoma-related features [44]. The sGA group, IPI score, and hemoglobin level (cut-off Hb = 12 gr/dL) were key prognostic features in building the elderly prognostic index (EPI). Interestingly, in developing the EPI, the prognostic weight of frailty status was higher than that of high-risk IPI (4 versus 3), thus confirming the relevant role of sGA in determining a patient’s outcome. Finally, three main risk groups were generated, each with a significantly different death probability (Figure 2A). The 3-year OS was 87%, 69%, and 42% for the low-, intermediate-, and high-risk groups, respectively. It is interesting to note that palliation was more frequently prescribed to patients in the intermediate and high-risk groups, who presented a mix of fitness status impairment and adverse lymphoma features. Interestingly, full- and reduced-dose therapeutic approaches were associated with similar OS in EPI intermediate- and high-risk patients, suggesting that the choice between full or reduced-dose regimens in older DLBCL patients can be made by balancing fitness status and disease characteristics (Figure 3).

A further strength of the EP study was the reproducibility of the EPI results, obtained with a validation cohort of 328 cases by pooling data from three different datasets (Figure 2B).

Utilizing this same EP cohort, an ad hoc analysis aimed to investigate the EPI score’s ability to predict the risk of early mortality in older DLBCL patients [48]. Sixty-nine of the overall 354 deaths were classified as premature deaths, with a cumulative incidence of 6% at three months and 10% in EPI high-risk patients. This study demonstrated that a considerable number of premature deaths are mainly associated with patient-related predictors. Moreover, a multivariable model indicated that high-risk EPI and bulky disease are independently associated with early mortality in patients treated with curative intent.

Since 10% of the entire EP registered cases were DLBCL patients over the age of 80, there is a growing interest in this population due to their improved general condition, leading Tucci to explore whether there are any substantial differences in therapeutic management and outcomes in these therapeutically difficult-to-approach cases [49]. Early octogenarians (EO, aged 80–84 years), representing one-third of patients over age 80, were compared with late octogenarians (LO, aged ≥85 years) and with those defined as unfit/frail on sGA enrolled in the EP. The 2-year OS and progression-free survival (PFS) rates were significantly higher in the EO group (2-year PFS rate, 56%; 2-year OS rate, 63%) compared with the LO group (2-year PFS rate, 43%; 2-year OS rate, 48%). Interestingly, the Kaplan–Maier curves of EO and unfit/frail cases <80 years overlapped. As expected, half of the LO patients were treated with palliation compared with 23% of EO cases, with a significantly lower 2-year OS rate (43% versus 56%, *p* = 0.01). Moreover, no difference was found in terms of 2-year OS between EO and LO patients receiving either full- or reduced-dose anthracycline. Rituximab added to palliative therapy significantly improved the outcome (2-year OS rate of 42% versus 22%). Finally, the EPI score and, to a lesser extent, sGA maintained their independent prognostic power even when considering only patients treated with anthracycline-containing regimens.

The main takeaways of this paper are that (i) a geriatric assessment before starting treatment in older patients with DLBCL should be considered mandatory, and the sGA is a useful and validated tool for this purpose, (ii) both full- and reduced-dose anthracycline regimens positively impact outcome measures, and (iii) the high-risk subgroup identified by EPI requires further investigations that also consider new monoclonal antibodies [11,12] and chemo-free approaches [50,51]. Ongoing trials include the sGA and EPI scores and will be available for further validation of these two indexes.

### 2.3. Retrospective Population-Based Study: The Simplified Frailty Score

A population-based cohort of 784 DLBCL patients aged ≥70 years identified through the cancer registry of Norway (CRN) was analyzed retrospectively [52].

Besides age, several candidate variables were used in their model: GA, ADL, CCI, body mass index (BMI), albumin, and the geriatric nutritional risk index (GNRI) scores [53]. The GNRI is an adaptation of the nutritional risk index for older subjects [54], with cutoffs (absent/low, moderate, and severe) prognostically validated in DLBCL [55]. All the above-mentioned items covered functional, nutrition, and comorbidity status. The final frailty model consisted of ADL, CCI, GNRI, and age ≥85 years and included all variables in a multivariable Cox regression analysis for OS in a training cohort of 522 cases. A significantly inverse relationship was found between the cumulative frailty score and survival rate. Finally, cases were clustered based on their cutoff values (1 for fit, 1.5–3 for unfit, and >3 for frail), showing the impressive prognostic power of frailty groups to predict OS, even when adjusted for other variables (IPI score, Ann Arbor stage, age group, and ECOG PS), and the analysis was restricted to patients receiving chemotherapy. In this respect, four therapy-intensity classes were considered based on the planned first cycle: full-dose R-CHOP, reduced-dose R-CHOP, anthracycline-free regimens, or palliation. Isaksen et al. [52] demonstrated better results with full-dose R-CHOP than with attenuated R-CHOP in terms of both OS and PFS in fit patients (2-year OS, 86% vs. 70%, *p* = 0.012; 2-year PFS, 85% vs. 63%, *p* = 0.002). Nevertheless, unfit patients attained overlapping benefits regardless of R-CHOP treatment intensity, suggesting that attenuated R-CHOP could be sufficient for these patients. Poor survival was confirmed for frail patients, although some benefitted from attenuated R-CHOP. Thus, the authors of this retrospective population-based study developed and validated a frailty score that convincingly predicted outcomes in patients treated with different degrees of dose intensity and suggested that this frailty score is particularly suitable for assisting physicians in selecting appropriate therapeutic algorithms in the daily management of older DLBCL patients.

### 2.4. Other Retrospective Studies

A retrospective GELTAMO study involving 252 DLBCL patients aged 80–100 years old diagnosed in 19 centers over 12 years was conducted to investigate the impact of treatment and GA scales (CIRS, CIRS-G, and CCI) on PFS and OS. A multivariable model indicated age <86, CIRS <6, intermediate-risk R-IPI, and anti-CD20-containing regimens as independent predictors of reduced progression and mortality risk [56]. A sub-analysis performed on chemotherapy-treated patients revealed a significantly reduced risk of progression or death for those cases treated with R-CHOP, either at full or reduced doses, with age <86 years, CIRS <6, and intermediate-risk R-IPI. Finally, a 2-cluster prognostic score (0–1 versus 2–3 risk factors) using age, CIRS, and R-IPI showed an independent association with outcomes.

Searching for less time-consuming and challenging methods to perform GA, Sakurai et al. focused on the usefulness of the Flemish version of the Triage Risk Screening Tool (fTRST) and the G8 in 59 older DLBCL patients [57]. The fTRST (five items related to cognitive impairment, solitary living, difficulty walking, hospitalizations, and polypharmacy) and the G8 (eight items related to food intake, weight loss, mobility, neuropsychological issues, body mass index, polypharmacy, self-perception of health, and age) geriatric tools are robust predictive indicators of older cancer patients’ outcomes [58]. Therefore, the authors conducted a single-institution retrospective study evaluating fTRST and G8, integrated with the CCI, to predict the prognosis of these older DLBCL patients. The 2-year OS rate remained significantly worse in both abnormal fTRST and G8, even after adjusting for other significant variables by multivariable analysis. However, the inadequately limited number of patients and the retrospective nature of the investigation limited the study’s strength.

Lee et al. [59] conducted a multicenter study aimed at substantiating the prognostic power of the G8 and the fTRST, retrospectively reviewing the electronic medical records for OS of 388 older DLBCL patients. In the multivariable COX model, the G8 as a continuous variable was an independent predictor of mortality. According to the ROC-derived cutoff value, a Kaplan–Meier overall survival analysis showed that the patients with a G8 score of ≥9.5 points had a significantly longer OS than those with a G8 score of <9.5 points, with an inverse linear relationship between the G8 score and mortality risk. Notably, the G8 cutoff value (9.5 points) was lower than that (cutoff = 14) reported in solid tumors [58]. Finally, the authors calculated that an improvement in OS could be reached by maintaining at least ≥80% of the average delivered relative dose intensity in the higher G8 (>9.5 points) group and ≥60% in the lower G8 (≤9.5 points) group, suggesting that the G8 score could be used to modify the chemotherapy dose of individual patients.

As part of the lymphoma epidemiology of outcomes (LEO) cohort study, the vulnerable elders survey (VES-13) was administered to 2004 indolent and high-grade non-Hodgkin lymphoma patients [46]. The VES-13 is a simple, self-reported, function-based tool whose items include the patient’s age, self-rated overall health status, functional limitations in physical activity, and functional disabilities in more complex activities of daily living. In the overall lymphoma subtype, vulnerable status predicted 1-year mortality in both older (≥65 years) (odds ratio (OR) =1.96; 95% CI 1.15–3.36; C-statistic 0.80) and young (OR = 6.38; 95% CI 3.07–13.3; C-statistic 0.83) accrued patients. Of these, 521 were DLBCL. VES-13 identified 38% of those patients with a vulnerable status, also predicting in this subset a 2-times higher risk of 1-year mortality after adjusting for age, PS, and disease stage [46].

### 2.5. New Tools to Assess the Health Status of the Elderly: The Case of Sarcopenia

Sarcopenia involves the loss of skeletal muscle mass and function, which commonly occurs with advancing age and several long-term conditions. Moreover, this disorder is associated with an increased likelihood of adverse outcomes, including falls followed by fractures, physical disability, and increased mortality risk. Acknowledging sarcopenia in clinical practice is relatively new but critical, considering the array of its unfavorable health effects. Although its original definition focused on muscle mass, an innovative description now highlights muscle function, as clarified in several international guidelines [60]. Several tools have been developed to determine sarcopenia status and physical performance. The revised European consensus recommends a specific algorithm for case discovery and severity determination in sarcopenic patients. Among the proposed tools for the initial screening for sarcopenia risk are the self-reported 5-item SARC-F (strength, assistance with walking, rising from a chair, climbing stairs, and falls) questionnaire as well as whole-body resonance imaging to evaluate muscle mass and quality.

Since several studies have reported ambiguous results concerning the association between sarcopenia and outcomes in DLBCL, a meta-analysis was conducted to assess the potential prognostic influence of sarcopenia on clinical outcomes in DLBCL patients undergoing standard treatment [61]. Interestingly, sarcopenia was more frequent in DLBCL patients than it was in the general population. Moreover, this meta-analysis of 11 published studies concluded that there was a significant association between poor sarcopenia status and worse survival, even after adjusting for other potential confounders. This may be due to the negative effect of sarcopenia on the probability of completing the planned therapy [60]. Thus, future prospective studies incorporating tools for sarcopenia screening and measurement are warranted.

## 3. Conclusions

In hematological trials developed for older patients, the primary focus has been on standard endpoints; patient-centered outcomes have been included in less than one-fifth of studies.

This review demonstrates that, regardless of the specific tools employed, a routine geriatric assessment of DLBCL patients before therapy starts must be performed. Indeed, not performing this assessment deprives physicians of an outstanding tool for predicting the risk of progression or death and for supporting therapeutic decision-making.

Moreover, GA can detect unidentified problems and risks, thereby suggesting which target interventions to prescribe.

In this context, sGA and EPI, validated in the large observational elderly project, seem to be excellent tools to standardize clinical practice and in future research on older DLBCL patients. An important feature of GA is that it should be as little time-consuming as possible to guarantee its routine use in clinical practice. From this perspective, sGA is an easy-to-use, reproducible tool that takes physicians under ten minutes to perform.

Nevertheless, sGA and all similar geriatric scale systems can be further improved. In this respect, sarcopenia is a relatively new element that has been increasingly examined in the prognostic evaluation of patients with DLBCL treated with curative intent. Thus, it could be incorporated into future prospective assessments. New tools, based on prospective studies, can help physicians determine the best treatment based on the individual patient’s characteristics. Finally, alongside the classical and evergreen R-CHOP, new therapeutic strategies, i.e., monoclonal antibodies and chemo-free approaches, are expanding the curative armamentarium of DLBCL. These novel therapies are critical to improving the outcomes of hard-to-treat older patients. In this regard, Figure 4 is the author’s own attempt to allocate new treatments to older patients assessed by EPI, sGA, and age. However, future large prospective trials are hopefully planned that have expanded eligibility criteria to include unfit and frail older patients and relevant endpoints for the elderly other than the ‘life expectancy’. Lastly, pragmatic clinical trials, conducted in the standard-of-care setting as an alternative to classic randomized trials, should be designed [62].

## Figures and Tables

**Figure 1 cancers-15-05845-f001:**
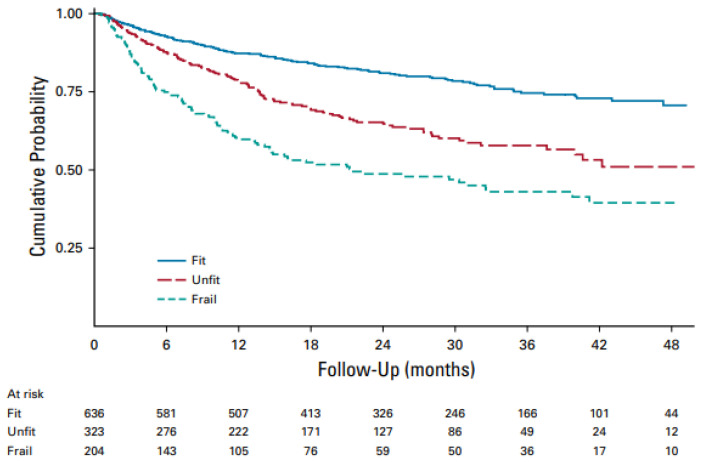
Overall survival by sGA in all patients with treatment details (n = 1163). Abbreviations: sGA, simplified geriatric assessment [44].

**Figure 2 cancers-15-05845-f002:**
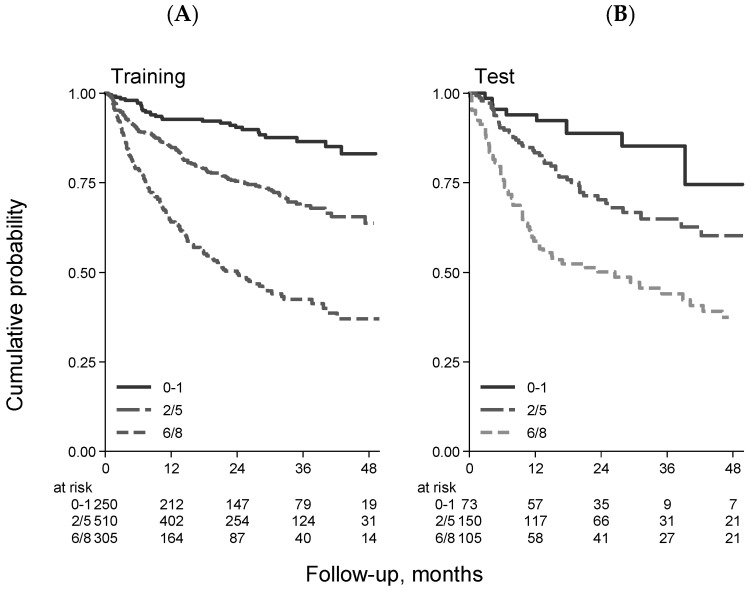
Overall survival stratified by EPI in the training ((**A**) 1065 patients) and validation ((**B**) 328 patients) samples [44].

**Figure 3 cancers-15-05845-f003:**
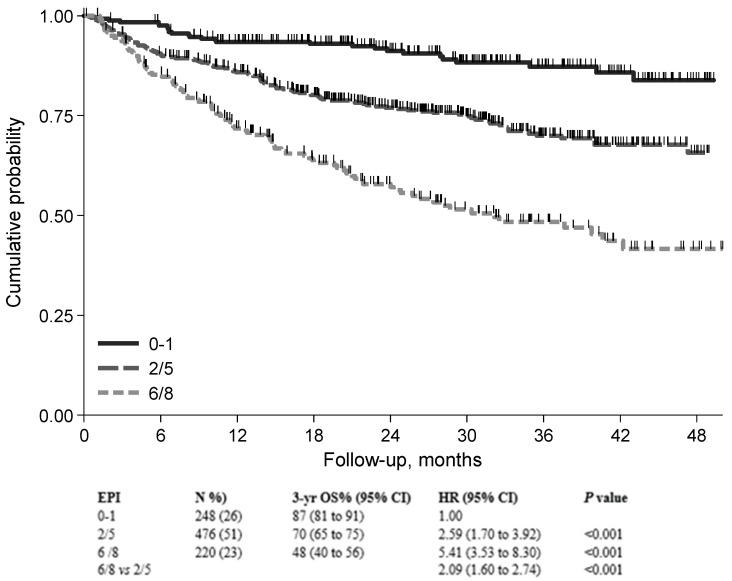
Overall survival for patients treated with anthracycline-containing regimen stratified by EPI risk group (n = 944). Abbreviations: EPI, elderly prognostic index [44].

**Figure 4 cancers-15-05845-f004:**
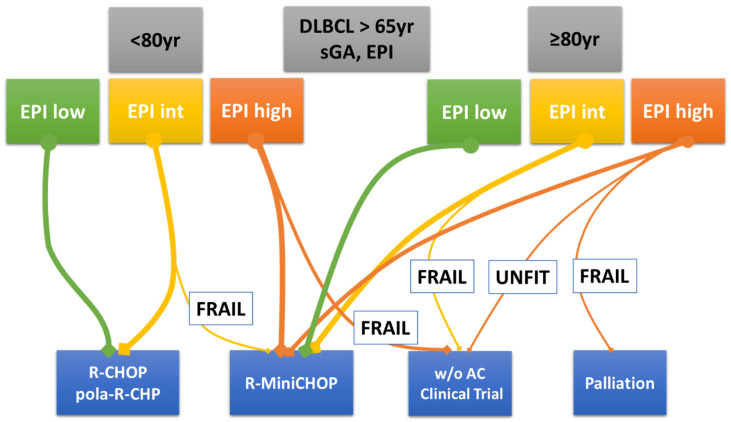
Author’s personal decision tree for older patients with DLBCL. Legend: The thicker the line, the more frequently that therapeutic option is adopted. Abbreviations: DLBCL, diffuse large B-cell lymphoma; sGA, simplified geriatric assessment; EPI, Elderly Prognostic Index; w/o AC, without anthracycline.

**Table 1 cancers-15-05845-t001:** Criteria for sGA assessment [44].

	FIT	UNFIT		FRAIL
ADL	≥5 *	<5 *	6 *	<6 *
	and	and/or	and	and/or
IADL	≥6 *	<6 *	8 *	<8 *
	and	and/or	and	and/or
CIRS-G	0 score = 3–4 and ≤8 score = 2	≥1 score = 3–4 and/or >8 score = 2	0 score = 3–4 and <5 score = 2	≥1 score = 3–4 and/or ≥5 score = 2
	and	and	and	and
Age	<80	<80	≥80	≥80

Abbreviations: ADL, activities of daily living; CIRS-G, Cumulative Illness Rating Scale for Geriatrics; IADL, instrumental ADL; sGA, simplified geriatric assessment. * Number of residual functions.

## Data Availability

Data are contained within the article.

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
