# Peer review of "The Role of Geriatric Assessment in the Management of Diffuse Large B-Cell Lymphoma"

_cancers, 2023, doi:10.3390/cancers15245845_

Round 1
Reviewer 1 Report
Comments and Suggestions for Authors
Nice desription of the development and evolution of geriatric assessments. interesting paper, well written and clinically very important.
Author Response
Thank you for your positive comment.
Reviewer 2 Report
Comments and Suggestions for Authors
The author submitted a review article entitled “The Role of Geriatric Assessment in the Management of Diffuse Large B-Cell Lymphoma” focusing on the Geriatric Assessment in the Management of Diffuse Large B-Cell Lymphoma (DLBCL) referring to their own index, Elderly Prognostic Index (EPI), and others. The manuscript is intriguing for the readers of “Cancers” because more elderly patients are suffering from DLBCL recently as described by the authors. However, there are several issues to be clarified.
Majors)
1) Line 5) “Haematology Unit Arcispedale S. Maria Nuova, AUSL-IRCCS, Reggio Emilia on behalf of Elderly Lymphoma 5 Committee of Fondazione Italiana Linfomi (FIL)”
“on behalf of Elderly Lymphoma 5 Committee of Fondazione Italiana Linfomi (FIL)” is better to be deleted because legend of Figure 4 is described as “Author’s personal decision tree for older patients with DLBCL”
Line 52) “With no validated, objective tools to evaluate the patient and the disease, the therapeutic decision is left to the subjective assessment of the physician, with a high and unacceptable risk of exposing the patient to unnecessary toxicity or to ineffective therapies.”
The sentence is better to be de-emphasized.
Minors)
1) Line 55) “In this article, we review the main tools that can be used to support the initial assessment of the older subject and how these tools can be used to foresee patient outcomes and to support treatment decisions.”
“patient outcomes” is better to be “DLBCL-patient outcomes”.
2) Table 1 and Figures 2, 3 is better to include each reference.
3) “EPI risk groups: Low: 0-1, Intermediate: 2-5, High: 6-8” in Figure 2 and Figure 3 are confusing.
Author Response
1) Line 5) “Haematology Unit Arcispedale S. Maria Nuova, AUSL-IRCCS, Reggio Emilia on behalf of Elderly Lymphoma 5 Committee of Fondazione Italiana Linfomi (FIL)”
“on behalf of Elderly Lymphoma 5 Committee of Fondazione Italiana Linfomi (FIL)” is better to be deleted because legend of Figure 4 is described as “Author’s personal decision tree for older patients with DLBCL”
WE AGREE WITH THIS COMMENT AND DECIDED TO MODIFIY THE AUTHORSHIP ADDING THE NEMAE OF THE COAUTHORS WHO ACTUALLY CONTROBUTED TO THE WRITING OF THIS MANUSCRIPT. THE NAMES AR ADDED TO THE REVISED MS.
Line 52) “With no validated, objective tools to evaluate the patient and the disease, the therapeutic decision is left to the subjective assessment of the physician, with a high and unacceptable risk of exposing the patient to unnecessary toxicity or to ineffective therapies.”
The sentence is better to be de-emphasized.
WE DE-EMPHASIZED THE SENTENCE AS SUGGESTED.
Minors)
1) Line 55) “In this article, we review the main tools that can be used to support the initial assessment of the older subject and how these tools can be used to foresee patient outcomes and to support treatment decisions.”
“patient outcomes” is better to be “DLBCL-patient outcomes”.
OK
2) Table 1 and Figures 2, 3 is better to include each reference.
OK
3) “EPI risk groups: Low: 0-1, Intermediate: 2-5, High: 6-8” in Figure 2 and Figure 3 are confusing.
OK
Reviewer 3 Report
Comments and Suggestions for Authors
Merli F discusses very ellegantly in this review the current state of frailty assessment in DLBCL. He covers all topics in a comprehensive manner.
I will only include in the references the following paper and make a comment in the discussion:
Hormigo-Sanchez AI, Lopez-Garcia A, Mahillo-Fernandez I, Askari E, Morillo D, Perez-Saez MA, Riesco M, Urrutia C, Martinez-Peromingo FJ, Cordoba R, Gonzalez-Montalvo JI. Frailty assessment to individualize treatment in older patients with lymphoma. Eur Geriatr Med. 2023 Oct 12. doi: 10.1007/s41999-023-00870-2.Epub ahead of print. PMID: 37823983.
It is the last paper published of a prospective GA in older patients with DLBCL highlighting the need of a tailor therapy.
Author Response
I will only include in the references the following paper and make a comment in the discussion:
Hormigo-Sanchez AI, Lopez-Garcia A, Mahillo-Fernandez I, Askari E, Morillo D, Perez-Saez MA, Riesco M, Urrutia C, Martinez-Peromingo FJ, Cordoba R, Gonzalez-Montalvo JI. Frailty assessment to individualize treatment in older patients with lymphoma. Eur Geriatr Med. 2023 Oct 12. doi: 10.1007/s41999-023-00870-2.Epub ahead of print. PMID: 37823983.
It is the last paper published of a prospective GA in older patients with DLBCL highlighting the need of a tailor therapy.
OK